# Long-Term (2002–2021) Trend in Nutrient-Related Pollution at Small Stratified Inland Estuaries, the Kishon SE Mediterranean Case

Barak Herut *, Yaron Gertner, Yael Segal, Guy Sisma-Ventura, Nurit Gordon, Natalia Belkin and Eyal Rahav *

Israel Oceanographic and Limnological Research (IOLR), National Institute of Oceanography,
Haifa 3108001, Israel
* Correspondence: barak@ocean.org.il (B.H.); eyal.rahav@ocean.org.il (E.R.)

**Abstract:** Nutrient pollution may negatively affect the water quality and ecological status of rivers and estuaries worldwide, specifically in stratified and small inland estuaries. We present a long-term, two-decade data set of dissolved inorganic nutrient concentrations, chlorophyll-*a* (chl-*a*), dissolved oxygen (DO), and potentially toxic algal cell concentrations at the Kishon River estuary (Israel) as a case study for assessing nutrient ecological thresholds in such type of estuaries, prevalent along the Mediterranean coast of Israel. In-situ measurements and water samples were collected at 3 permanent stations at the lower part of the estuary every March and October/November in 40 campaigns over the years 2002 to 2021. In spite of an improvement in nutrient loads and concentrations as recorded over the last 2 decades, the nutrient and chl-*a* levels at the Kishon estuary surface water represent mostly a 'bad' or 'moderate' ecological state, considering the recommended thresholds discussed in this study. It is suggested to develop a combined suite of nutrient and biological variables for assessing Good Environmental Status (GES), considering the relatively high residence time of such small, low-flow estuarine water bodies.

**Keywords:** nutrients; river; estuary; water quality; pollution; seawater; Mediterranean





## 1. Introduction

Coastal areas worldwide, and in the Mediterranean in particular, have experienced a high population increase in the last decades [1], leading to higher water demands. It is projected that due to climate change, freshwater availability is likely to decrease substantially [2], while already in limited availability per capita (stress to scarcity) for most countries in the southern and eastern part of the Mediterranean basin [1,2], demanding water quality protection of water resources. The increased water demands for drinking, irrigation/food security, and manufacturing may also further deteriorate the water quality and coastal ecosystems. Specifically, anthropogenic activities have resulted in multiple pressures on freshwater resources requesting their protection and sustainable use via national and international legislations, which include ecological criteria for good environmental status such as the Water Framework Directive (WFD) (2000/60/EC); the Clean Water Act via the US Environmental Protection Agency (EPA) guidelines) and assessment methodologies [3].

Elevated concentrations of nutrients are a major factor contributing to the failure of many water bodies to achieve good ecological status (GES). Recent assessments grade nutrient pollution as a major contributor to multiple pressures for water quality degradation in rivers and estuaries [4–6]. Nutrient pollution from both diffuse and point sources acts as a major contributor to the deprivation of European water bodies [4] and represents a relatively greater threat to water quality as compared to the 1970s in rivers in the USA [6]. Eutrophication due to excessive anthropogenic nutrient loading was documented worldwide in estuaries and coastal waters, which may also be associated with harmful algal blooms (HABs) [7,8].

The projected decrease in freshwater fluxes in the Mediterranean basin attributed to drier and warmer conditions related to climate change [2,9] superimposed on the population increase and anthropogenic pressures, will further impact the water quality, especially at the lower reaches of small coastal rivers. In the Southeastern Mediterranean (excluding the Nile River), small coastal rivers account for most of the drainage area. Several arid and semi-arid coastal areas in the Mediterranean contain low-stream estuaries with low base flows [10], as in the case along the ~190 km Israeli coast [11,12].

Along the Israeli Mediterranean shoreline at the SE Mediterranean Sea, the bathymetry of the lower parts of several coastal streams enables the penetration of seawater and the formation of ecologically unique highly stratified small-size estuaries up to a few kilometers inland [13–18]. These small estuarine water systems are exposed to severe anthropogenic pressures and low natural water flow, holding high nutrient contamination and low water quality conditions, and may induce HABs [19,20].

In this paper, we use a long-term, two-decade (2002–2021) dataset of dissolved inorganic nutrient concentrations, chlorophyll-*a* (chl-*a*), dissolved oxygen (DO), and potentially toxic algal species counting (including HAB events) at the Kishon River estuary (Israel) as a case study for assessing water quality trends in small stratified inland estuaries. Here we assess the Kishon River estuary ecological state by applying relatively robust nutrient thresholds recently evaluated for European (EU) rivers by Poikane et al. [21] and for chl-*a* and DO by the National Oceanographic and Atmospheric Administration (NOAA) of the U.S. [22]. For the most seaward station, we consider also the nutrient threshold recently suggested for coastal water at Haifa Bay [23], which is considered oligotrophic, as other coastal areas in the Levantine basin.

## 2. Materials and Methods

### 2.1. Study Site

The Kishon river drainage basin occupies approximately 1100 km$^2$, with intensive agricultural activity taking place within its drainage basin. The Kishon outflow enters Haifa Bay located in the southeastern Mediterranean Sea (Figure 1A), after passing through two harbors (Kishon and Haifa harbors). It is characterized, as many other Israeli coastal rivers outflowing the southeastern Mediterranean Sea, by low base flows, with discharges varying between 0.02 and 0.2 m$^3$ s$^{-1}$ in the summer and between 0.2 to 0.6 m$^3$ s$^{-1}$ in the winter [17]. The 7-km long estuary is exposed to the penetration of seawater because its bathymetry lies below sea level, thereby producing a highly stratified water column. The total annual discharge ranges between $3.8 \times 10^6$ and $1 \times 10^8$ cubic meters (Figure 1B), with a three decades annual average of $30.3 \times 10^6$ cubic meters (Israeli Water Authority Hydrological Reports). Along the estuary, there are two industrial plants producing fertilizers, an oil refinery, a sewage treatment plant [13], and additional smaller industries (Figure 1).

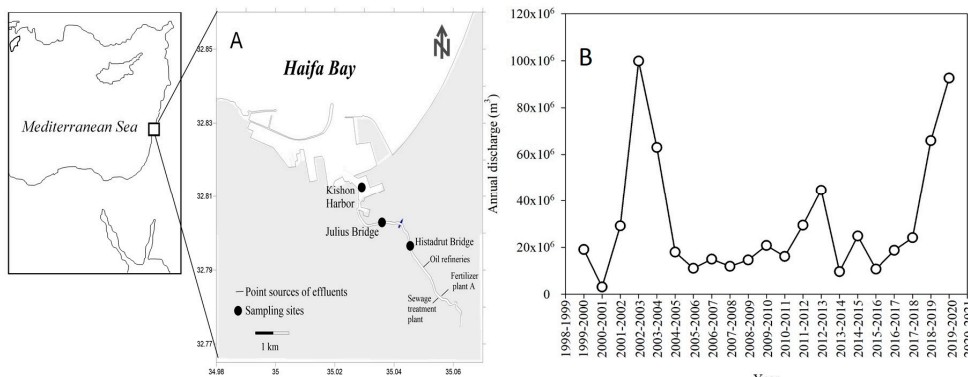

**Figure 1.** A map showing the sampling locations (Kishon Harbor; Julius Bridge; Histadrut Bridge, both ~1.4 and ~2.9 km upstream of the harbor station, respectively) at the lower reach of the Kishon River estuary between 2002–2021 (**A**), and the annual discharges at the estuary (data source: Israeli Water Authority Hydrological Reports) (**B**).

The outflow of the Kishon influences the water quality in southern Haifa bay, introducing nutrient-rich waters [15,24]. It thus affects the spatial distribution of nutrients in the bay and the enhancement of the primary-producers biomass, while fading off drastically seaward to oligotrophic coastal conditions [24–26].

### 2.2. Sampling

Surface water samples were collected at 3 permanent stations along the Kishon Estuary (Figure 1) every May and October/November in 40 campaigns over the years 2002 to 2021, thereby representing spring and autumn variability. These sampling locations represent the lower 2 km of the inland estuary of the river. At each sampling station, the water samples were collected by a horizontal Niskin bottle (General Oceanics, Miami, FL, USA) from ~10 cm below the surface. Measurements of salinity, dissolved oxygen (DO), temperature, and pH were taken in situ (~10 cm depth) using YSI 6600 probe (Yellow Springs, OH, USA) and PC6600 software or by Hydrolab-MS5 Multiparameter Sensor (Hach Environmental, Loveland, CO, USA). The pH, salinity, and dissolved oxygen sensors were calibrated before use.

### 2.3. Inorganic Nutrients

Pre-filtered (0.45 μm) water samples were transferred into acid-clean 20 mL plastic vials (triplicates) and were kept frozen until analysis ($-20$ °C). Nutrients were determined using a segmented flow Skalar SAN plus System Instrument [27] or with a Seal Analytical AA-3 system [28]. The limit of detection (LOD), estimated as two times the standard deviation of 10 measurements of the blank (low nutrient aged seawater collected from the off-shore surface at the Levantine basin) for $PO_4^{3-}$, $Si(OH)_4$, $NO_2^- + NO_3^-$ (NOx) and $NH_4^+$ were 9, 50, 80 and 90 nM, respectively. The reproducibility of the analyses was determined using certified reference materials (CRM): MOOS 3 ($PO_4^{3-}$, $NO_X$, and $Si(OH)_4$), VKI 4.1 (NOx and $NH_4^+$), and VKI 4.2 ($PO_4^{3-}$ and $Si(OH)_4$). The sample analysis results were accepted when measured CRMs were within $\pm10\%$ of the certified values.

### 2.4. Chlorophyll-a Extraction

Water samples (50–100 mL) were pre-filtered through 63 μm mesh and then onto GF/F filters (nominal pore size ~0.7 μm), wrapped in aluminum foil, and frozen ($-20$ °C). Chlorophyll-*a* (Chl-*a*) pigment was extracted from the filters using acetone (90%) overnight and determined by the non-acidification method [29] using a Turner Designs (Trilogy) fluorimeter (USA).

### 2.5. Harmful Microalgae Identification

Potentially toxic microalgae were identified and quantified by epifluorescence and bright-field microscopy. Samples (up to 300 mL) were concentrated with a 3 μm polycarbonate filter using the filter-transfer-freeze (FTF) method [30]. The filters were then transferred to microscope slides, with the filtered cells face down in a drop of water, and allowed to freeze. After peeling the filters off the frozen slide, the cells were covered by a thin layer of glycerin gel, a glycerol drop, and a cover slide. Cell identification and enumeration were carried out by using an epifluorescence microscope (Olympus BX51, Evident Corporation, Tokyo, Japan) equipped with a chlorophyll filter (Ex: 450 nm, Em: 680 nm), at a magnification of $\times40$.

### 2.6. Statistical Tests

The temporal trends of dissolved nutrient concentrations and their endmembers, potentially toxic algae cell abundance, and chl-*a* concentrations were evaluated using a Pearson correlation test with a confidence level of 95% ($\alpha = 0.05$). Statistical tests were run using XLSTAT (New York).

### 2.7. Calculating the Nutrient Anthropogenic End-Member

Considering a conservative mixing along the salinity gradient of the 3 stations (see Section 3), we modeled the nutrient ($NO_2^-$ + $NO_3^-$, $NH_4^+$, and $PO_4^{3-}$) concentrations of the freshwater endmember in the most upstream Histadrut station for each sampling campaign, using the following equations:

$$\text{Nut-riv} = (\text{Nut-fw} \times f) + [\text{Nut-sw} \times (1 - f)] \tag{1}$$

where Nut-fw and Nut-sw designate the freshwater and seawater endmember nutrient concentrations, respectively, Nut-riv the river surface water nutrient concentration and f is the fraction of the freshwater endmember calculated in terms of the salinity:

$$f = \frac{\text{Sriv} - \text{Ssw}}{\text{Sfw} - \text{Ssw}} \tag{2}$$

where Sfw and Ssw designate the freshwater and seawater salinity endmembers, considered here as 5 and 40 psu, respectively, and Sriv is the measured river surface water salinity.

The nutrient concentrations of the seawater endmember (Nut-sw) used here represent the reference nutrient levels during the spring at the shallow water in Haifa Bay as detailed in Kress et al. [20]: $NO_2^-$ + $NO_3^-$–1.4 ± 0.7 µM; $NH_4^+$–0.52 ± 0.3 µM; $PO_4^{3-}$—0.048 ± 0.015 µM.

By combining Equations (1) and (2) we calculated the Nut-fw value for each sampling campaign at the Histadrut station by:

$$\text{Nut-fw} = [-35 \times (\text{Nut-riv} - \text{Nut-sw})]/[(\text{S-riv} - 40) + \text{Nut-sw}] \tag{3}$$

## 3. Results and Discussion

### 3.1. Nutrients, chl-a, and Dissolve Oxygen Levels

The variability of dissolved inorganic nutrients, DO and chl-*a* concentrations in surface water during 2002–2021 are presented as box plots in Figure 2 and Table 1. While a relatively large range of concentrations is presented with some remarkable peaks/lows, consistently significantly higher nutrient levels were observed at the most upstream (Histadrut) station (Figure 2), presenting the lowest salinities (Table 1). Consequently, the Histadrut station recorded the highest chl-*a* concentrations (Figure 2) and hence algal biomass load, impacting its bottom water's low (hypoxic-anoxic) DO concentrations and high microbial heterotrophic activity [18,20]. The DO levels in surface water were similar at all stations, while a few low DO concentrations were measured at the Histadrut and Julius stations (Figure 2). The pH at the surface water ranged between seawater (~8.1) and ~7.9 at the upstream station (Table 1). The observed scattering (standard deviations; lowest and highest quartiles and data points; Figure 2 and Table 1) of the nutrient and chl-*a* concentrations decrease downstream, displaying the highest variability upstream at the Histadrut station. The latter may reflect fluctuations in the anthropogenic discharges, while the overall variability is also attributed to seasonal imprints.

**Table 1.** Summary table showing the main surface water physiochemical and chl-*a* characteristics at the Kishon Estuary sampling stations between 2002–2021. The data shown are the averages and their corresponding standard deviation. DIN = $NO_3^-$ + $NO_2^-$ + $NH_4^+$.

| Variable | Harbor | Julius | Histadrut |
|---|---|---|---|
| Temperature (°C) | 24.1 ± 1.8 | 24.5 ± 2.0 | 24.4 ± 2.2 |
| Salinity (psu) | 35.5 ± 4.3 | 21.3 ± 7.2 | 17.8 ± 6.1 |
| DO (mg L$^{-1}$) | 8.3 ± 2.4 | 8.6 ± 4.4 | 7.6 ± 4.5 |
| pH | 8.1 ± 0.4 | 8.0 ± 0.3 | 7.9 ± 0.3 |
| $NO_2^-$ + $NO_3^-$ (µmol L$^{-1}$) | 104.5 ± 102.1 | 475.8 ± 267.2 | 701.0 ± 322.2 |
| $NH_4^+$ (µmol L$^{-1}$) | 23.0 ± 9.8 | 79.3 ± 8.7 | 137.4 ± 116.5 |
| $PO_4^{3-}$ (µmol L$^{-1}$) | 3.9 ± 2.8 | 10.6 ± 9.2 | 15.5 ± 15.2 |

**Table 1.** *Cont.*

| Variable | Harbor | Julius | Histadrut |
|---|---|---|---|
| DIN:PO$_4^{3-}$ | 119:1 | 91:1 | 81:1 |
| Si(OH)$_4$ (µmol L$^{-1}$) | 31.8 ± 32.7 | 133.4 ± 75.8 | 185.7 ± 107.4 |
| Chl-*a* (µg L$^{-1}$) | 21.1 ± 32.3 | 54.0 ± 7.0 | 72.1 ± 78.6 |

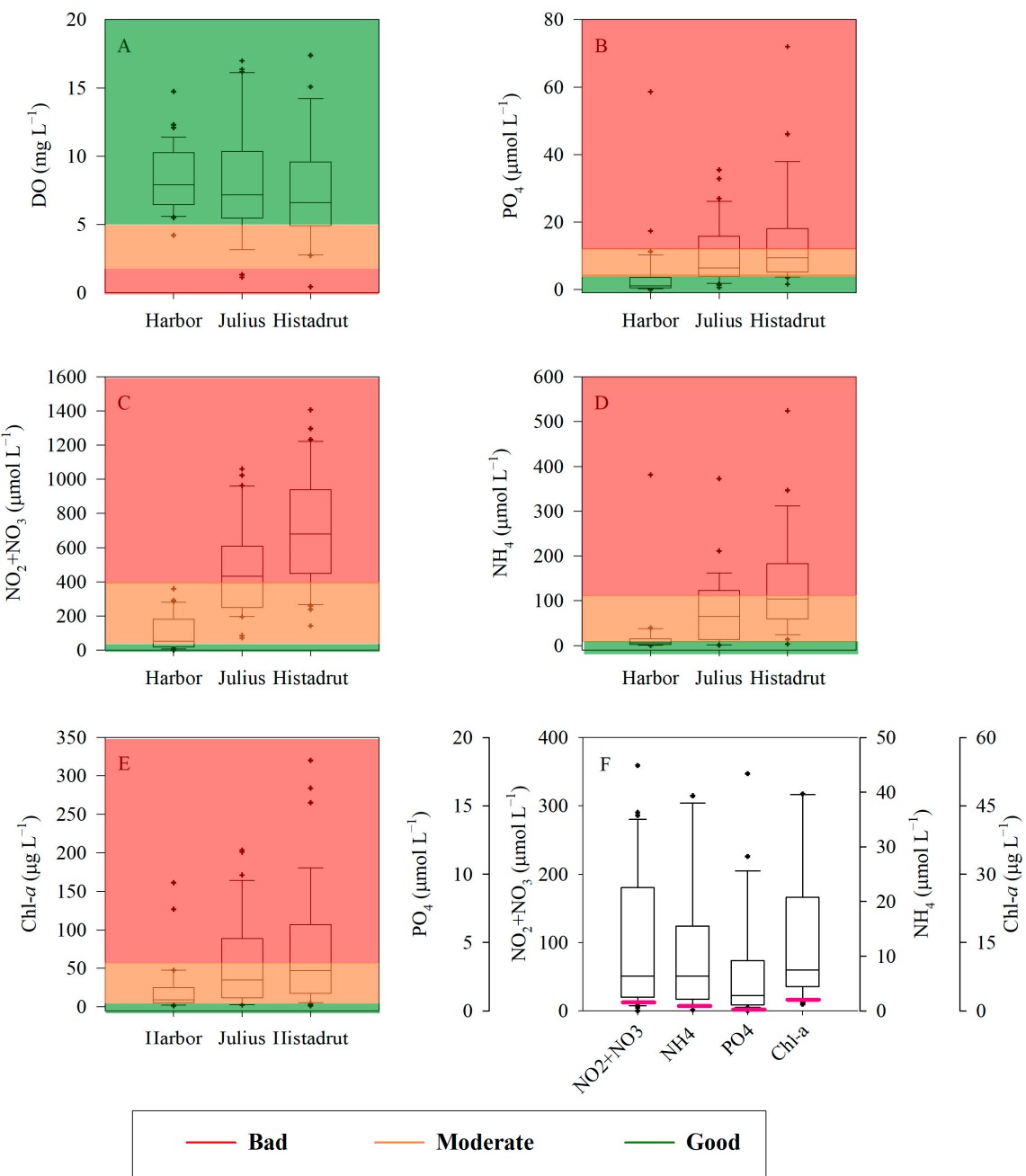

**Figure 2.** Whisker box plots showing the distribution of DO (**A**), PO$_4^{3-}$ (**B**), NO$_2^-$ + NO$_3^-$ (**C**), NH$_4^+$ (**D**), and chl-*a* (**E**) in the surface water of the Kishon Estuary between 2002–2021. Background colors represent the water quality thresholds as suggested here for small stratified inland estuaries (see Table 2), where red stands for bad water quality, orange for moderate, and green for the good ecological state. The nutrients and chl-*a* threshold between good ecological status (GES) and non-GES for Haifa Bay coastal water (SE Mediterranean Sea) derived from Kress et al. (2019) [23] are shown as pink lines within the box plots generated for the harbor station (**F**).

In most cases, a linear (or close) relationship exists between the nutrient concentrations and salinity per each sampling campaign, representing a conservative mixing curve between high-nutrient fresh/brackish water (hereafter 'freshwater') polluted endmember and a low-nutrient seawater endmember (Figure 3 for $NO_2^- + NO_3^-$). In some sampling campaigns deviation from a linear/conservative curve was observed, usually as a curve with a parabolic component, likely due to the removal of $NO_2^- + NO_3^-$ by biological processes along the path between the stations. Thus, a significant process impacting the nutrient concentrations in such estuary exposed to seawater intrusion is the degree of mixing between the anthropogenic freshwater endmember and seawater endmember with relatively low nutrient concentrations. Assuming a conservative mixing processes the $NO_2^- + NO_3^-$, $NH_4^+$, and $PO_4^{3-}$ endmember ranges may be assessed. The slope of the line is controlled mainly by the level of enrichment of nutrients in the polluted freshwater end member, and the degree of mixing with seawater. The polluted fresh endmember is affected by the level of discharge of treated industrial effluents enriched with nutrients into the river estuary and the concentrations at the upstream flow, which drain large agricultural areas and hence are impacted by fertilizers. Thus, the level of nutrient enrichment/pollution of the freshwater endmember is variable and is attributed to the changes in the effluents and the upstream discharges. This variability is exemplified by the measured distribution of $NO_2^- + NO_3^-$ concentrations versus salinity in the sampling campaigns (Figure 3). We can roughly estimate that the upper and lower slopes in Figure 3 represent the range of the highest and lowest $NO_2^- + NO_3^-$ levels in the freshwater end member. Considering a freshwater salinity endmember of 5 psu and the upper and lower linear trends ($NO_2^- + NO_3^- = -93 \times$ Salinity + 3620 and $NO_2^- + NO_3^- = -11 \times$ Salinity + 420, respectively), the $NO_2^- + NO_3^-$ endmember varies between approximately 360 to 3150 μM. The lowest points shown in the graph (marked as green triangles) are biased from all others, likely due to the top of the fertilizers plant during a prolonged worker strike in the fall of 2017, and hence the discharge of its nutrient effluents.

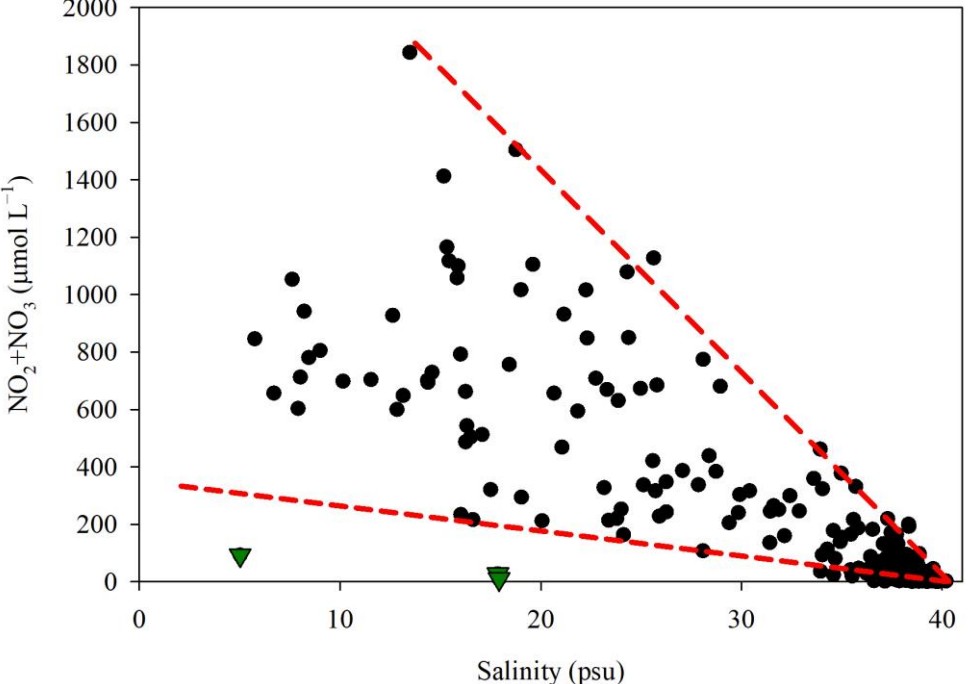

**Figure 3.** Relationship between $NO_2^- + NO_3^-$ concentrations and salinity in surface water at three stations along the Kishon lower estuary between 2002–2021. The bottom green triangles represent measured values at a period of almost no anthropogenic discharge (November 2017) due to a strike at the fertilizer industrial plant. The red lines represent mixing curves between the highest and lowest $NO_2^- + NO_3^-$ levels in the fresh/brackish water endmember and the seawater endmember (salinity ~40 psu).

### 3.2. Assessing the Nutrient Variations and Anthropogenic End-Member Dynamic

Applying Equation (3) and considering the Nut-sw concentrations as detailed above and the measured Sriv values, we calculated the Nut-fw value for each sampling campaign at the Histadrut station. Figure 4 presents the modeled nutrient concentrations of the freshwater end member and their variability with time. The calculated Nut-fw shows high annual variability, but a general decreasing trend for $NO_2^- + NO_3^-$ and $NH_4^+$ between 2002 and 2018, while $NO_2^- + NO_3^-$ and $PO_4^{3-}$ concentrations increases since 2019 (Figure 4A,C,E). The annual $NO_2^- + NO_3^-$, $NH_4^{+,}$ and $PO_4^{3-}$ concentrations at the surface water of the Histadrut station were most affected by the anthropogenic discharge of nutrients (Figure 4B,D,F). The relatively large observed variability is attributed to the fluctuating anthropogenic load of nitrogen and phosphorous (Figure 4A,C), variations in freshwater flow [17], and the intensity of seawater penetration upstream. A general decreasing (improving) trend in $NO_2^- + NO_3^-$, $PO_4^{3-}$ and $NH_4^+$ (despite some missing values of $NH_4^+$ in 2017–2021) was observed for the years 2002- 2018, which coincides with the general decreasing trend in the total N and P loads (Israel Ministry of Environmental Protection; https://www.gov.il/he/departments/guides/kishon_stream_point_pollution_loads?chapterIndex=5, accessed on December 2022) (Figure 4A,C). The loads were calculated using the annual cumulative effluent discharge per point source multiplied by the annual average N and P concentrations. Nonetheless, an unknown increase is observed in $PO_4^{3-}$ and $NO_2^- + NO_3^-$ concentrations measured since 2019–2021 that do not coincide with the reported anthropogenic N and P decreasing loads into the Kishon River from point sources. The observed increase may be a consequence of anthropogenic discharge from an unknown upstream location or hydrographic changes at the estuary attributed to the construction of a new Bayport.

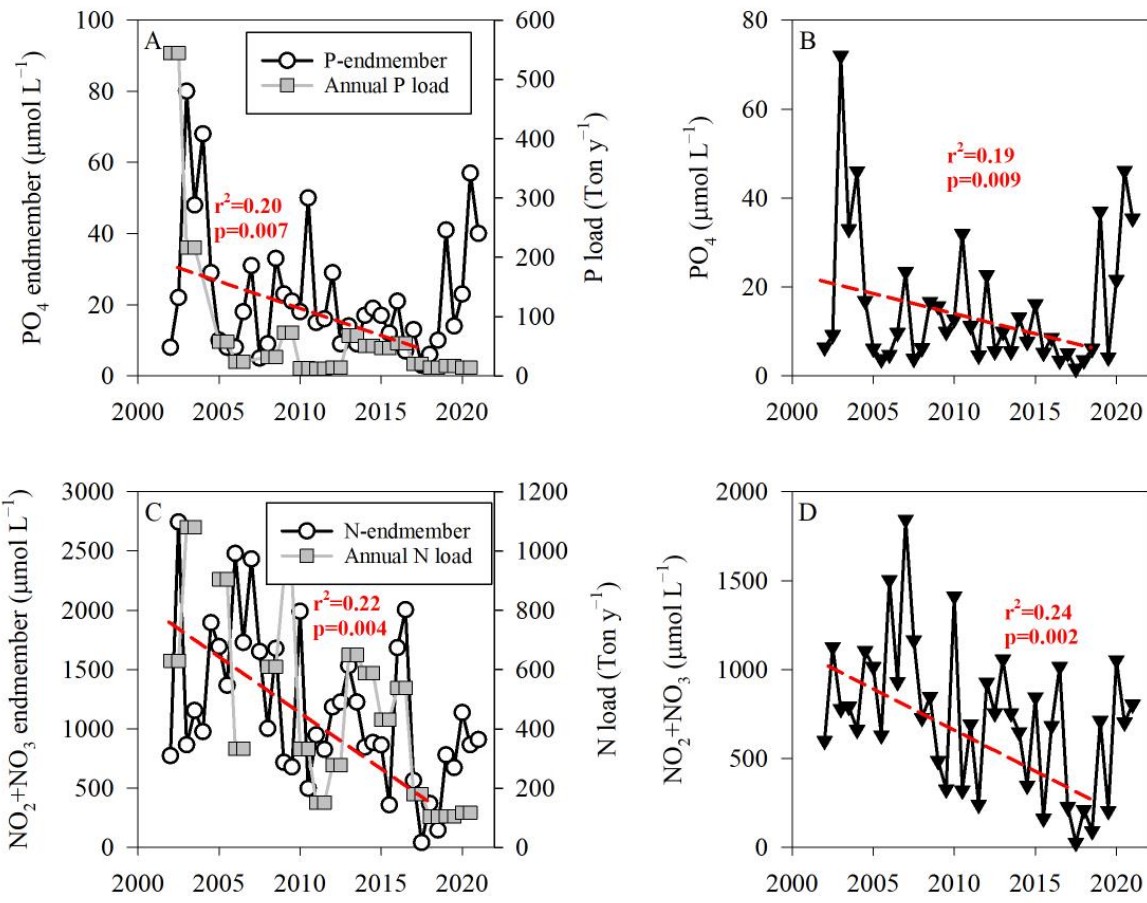

**Figure 4.** *Cont.*

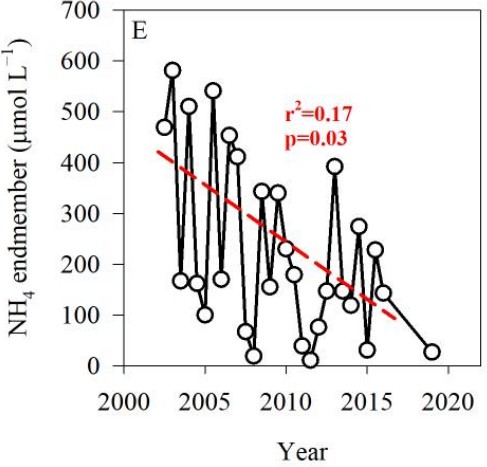
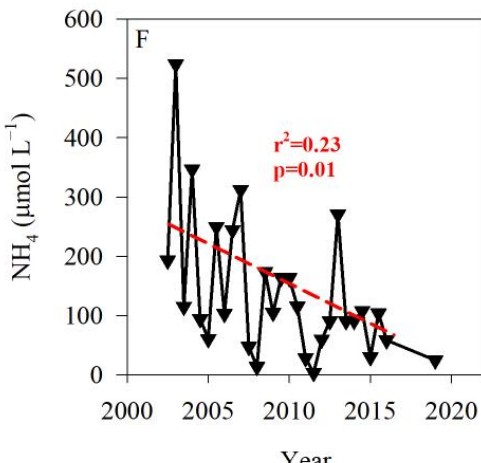

**Figure 4.** Time series of the calculated anthropogenic freshwater nutrient endmembers using Equation (3) ((**A,C,E**) circles) and the time series of the measured nutrient concentrations at the surface water of the Histadrut station ((**B,D,F**), triangles). Data are shown for $PO_4^{3-}$ (**A,B**), $NO_2^- + NO_3^-$ (**C,D**), and $NH_4^+$ (**E,F**). The total N and P loads into the Kishon river from anthropogenic point sources are presented as well (source: Israel Ministry of Environmental Protection, 2021 Report on pollutants budget; www.gov.il/he/departments/guides/kishon_stream_point_pollution_loads?chapterIndex=5, accessed on December 2022). The loads were calculated using the annual cumulative effluent discharge per point source multiplied by the annual average N and P concentrations. Note the different Y axis. The red lines show the linear decreasing trends for the period 2002–2018.

During the last 3 decades, mainly between 1994 and 2002, a drastic decrease of about 90% in the anthropogenic total N and P loads into the Kishon river estuary was reported (Israel Ministry of Environmental Protection; https://www.gov.il/he/departments/guides/kishon_stream_point_pollution_loads?chapterIndex=5, accessed on December 2022). The decrease in nutrient loads was associated with a change in the observed $NO_2^- + NO_3^-/PO_4^{3-}$ ratio at the Kishon River outlet (Kishon harbor) and southern Haifa Bay. Figure 5 presents the relationship between $NO_2^- + NO_3^-$ and $PO_4^{3-}$ concentrations at the Kishon harbor and southern Haifa Bay during this study, emphasizing a significantly higher $NO_2^- + NO_3^-/PO_4^{3-}$ ratio (average ~17:1) during 2002–2021 as compared to the reported values in 1993–2000 [15,24] of <5:1. Considering the change in the nutrient's N:P ratio, it is speculated that phytoplankton and bacteria in the southern Haifa Bay shifted from an N-limited (prior 2000) to P-limited system since 2002.

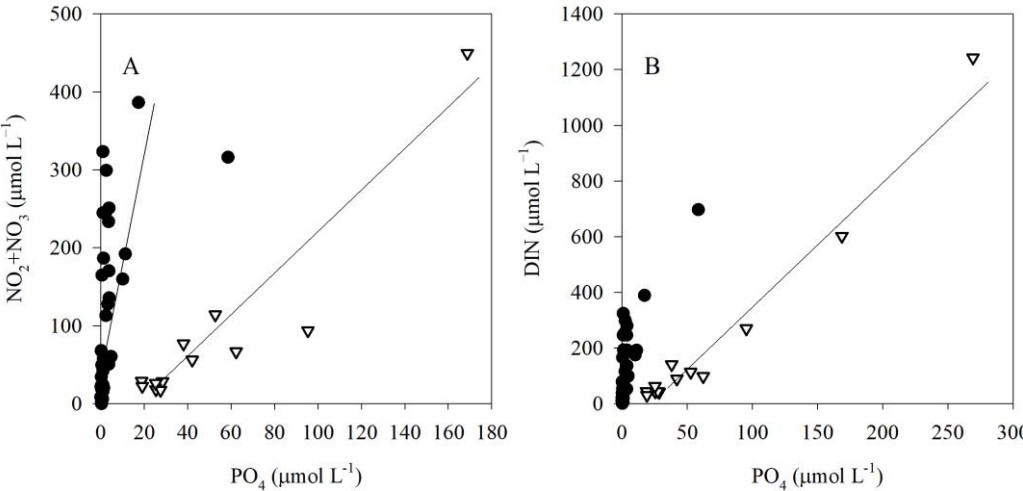

**Figure 5.** *Cont*.

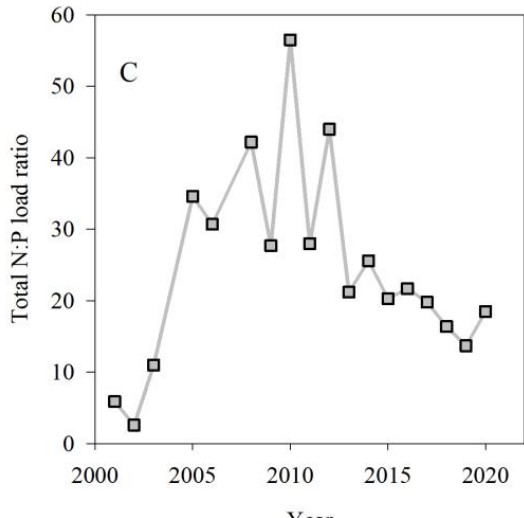

**Figure 5.** Relationship between $NO_2^- + NO_3^-$ (**A**) or DIN (**B**) vs. $PO_4^{3-}$ concentrations in surface water at the Kishon harbor during this study, 2002–2021 (black circle) and measurements performed during 1993–2000 (white triangles, [24,25]). The linear regression slopes for this study and 1993–2000 in panel A are 17.3 and 2.6, and in panel B are 11.3 (or 17.6 excluding the high anomalous value) and 4.6, respectively. The anthropogenic N:P load ratio is shown in panel (**C**) (grey square, calculated from the source: Israel Ministry of Environmental Protection, 2021 Report on pollutants budget; www.gov.il/he/departments/guides/kishon_stream_point_pollution_loads?chapterIndex=5).

### 3.3. Assessment of Water Quality Status

Many approaches were published for assessing the nutrient criteria in rivers and transitional water bodies aimed at supporting good ecological status (GES), with large criteria variations, recently evaluated for European (EU) waters ([5,21,22] and Table 2). The main issues contributing to this variability are related to the use of different nutrient species for different water categories/bodies, the use of different statistical parameters and approaches; definition of different nutrient thresholds for GES while considering different types of water within the main categories of rivers, transitional and coastal waters [21]. In addition, a generic typology for European (EU) rivers (and lakes) was suggested enabling more large-scale assessments across country borders and linking the water body types to habitat types under the European Habitats Directive [31]. An additional approach for deriving nutrient criteria and thresholds for different ecological statuses is combining nutrient measurements and biological quality elements (BQEs) for transitional and coastal waters presented by Salas Herrero F. et al. [32]. Most recently Nikolaidis N.P. et al. [33] assessed the threshold target concentrations of nutrients between good and moderate ecological status for EU rivers ranging from 35.7–250 μmol L$^{-1}$ total nitrogen (TN) and 0.35–3.5 μmol L$^{-1}$ total phosphorous (TP). Table 2 presents selected thresholds of nutrient criteria, mainly for the boundaries between the 'bad', 'moderate', and 'good' ecological state of rivers and transitional waters. These thresholds are based on different approaches that are not discussed here. In this study, the TP and TN were not monitored consistently, and therefore we used the species of dissolved inorganic nitrogen (i.e., $NO_2^- + NO_3^-$, $NH_4^+$) and soluble reactive phosphate (i.e., SRP or $PO_4^{3-}$) for the nutrient criteria, as well as chl-*a* and DO as complementary biological quality parameters. We adopted relatively robust thresholds considering the median and lower limits of the studies presented in Table 2. With regard to the Kishon harbor station, representing the most seaward part of the estuary and occupied primarily by seawater (and thus considered as a transitional water body), we considered also the coastal water threshold for GES as suggested by Kress et al. [23] (Table 2, Figure 2F).

**Table 2.** Summary of nutrient, chl-*a* and DO thresholds for river and estuarine water.

| Criteria/Status | Reference | PO$_4^{3-}$ (μM) | NO$_3$ (μM) | NH$_4^+$ (μM) | Chl-*a* (μg L$^{-1}$) | DO (mg L$^{-1}$) | TP (μM) | TN (μM) |
|---|---|---|---|---|---|---|---|---|
| Moderate | Poikane et al. [21]; median values | >2.6 | >143 | - | - | - | >4.8 | >164 |
| Good | medium-small rivers | ≤2.6 | ≤143 | - | - | - | ≤4.8 | ≤164 |
| Bad | | - | - | - | ≥60 | ≤2 | ≥3 | >70 |
| Moderate | Bricker et al. [22]; NOAA | - | - | - | 5–60 | 2–5 | 0.3–3 | 7–70 |
| Good | | - | - | - | ≤5 | ≥5 | ≤0.3 | ≤7 |
| Moderate/good target threshold | Nikolaidis et al. [32]; mid range | - | - | - | - | - | –1 | 57 |
| Bad | | >21 | >806 | >277 | - | <3 | - | - |
| Poor | | ≤21 | ≤806 | ≤277 | - | ≥3 | - | - |
| Moderate | Romero et al. [34] | ≤10.5 | ≤403 | ≤111 | - | ≥4 | - | - |
| Good | | ≤5.3 | ≤161 | ≤27.8 | - | ≥6 | - | - |
| High | | ≤1.1 | ≤32.3 | ≤5.6 | - | ≥8 | - | - |
| **Bad** | | **>10** | **>400** | **>110** | **>60** | **≤2** | **-** | **-** |
| **Moderate** | | **2.2–10** | **71–400** | **28–110** | **5–60** | **2–5** | **-** | **-** |
| **Good** | | **<2.2** | **<71** | **<28** | **<5** | **≥5** | **-** | **-** |
| | | | | Haifa Bay coastal seawater | | | | |
| GES/non-GES threshold | Kress et al. [23] | 0.1 | 2.1 | 0.78 | 1.14 | - | 0.13 | 12.7 |

At the Histadrut and Julius stations, approximately 90–97% of the $PO_4^{3-}$, 100% of the $NO_2^- + NO_3^-$, and 77–87% of the $NH_4^+$ measurements were classified as 'bad' or 'moderate' ecological state considering the river water nutrient criteria as adopted here (Figure 2 and Table 3). The Kishon harbor represents a transition zone between the river and the coast, therefore, based on its nutrient criteria it may be considered either as 'good' (Figure 2F and [23]) or as 'bad' (Figure 2; Table 3) ecological state, depending on the criteria used. The chl-*a* levels at the Histadrut and Julius stations reflect a 'bad' or 'moderate' ecological state for 85–90% of the measurements, and 76% for the Kishon harbor, typically as a moderate state (Figure 2). The non-GES at the Kishon harbor station based on the chl-*a* levels, despite the relatively 'good' nutrient state, if applying the river criteria in Table 2, strengthen the need to apply more strict coastal nutrient criteria for such transient water bodies, considering the coastal seawater criteria.

**Table 3.** The percent of nutrients and chl-*a* measured along the Kishon River Estuary in 2002–2021 falls within each of the status categories adopted in this study (Table 2).

| Variable | Station | Good | Moderate | Bad |
|---|---|---|---|---|
| $PO_4^{3-}$ | Histadrut | 3 | 53 | 44 |
| | Julius | 10 | 44 | 46 |
| | Harbor | 63 | 26 | 11 |
| $NO_3^-$ | Histadrut | 0 | 20 | 80 |
| | Julius | 0 | 43 | 57 |
| | Harbor | 44 | 56 | 0 |
| $NH_4^+$ | Histadrut | 13 | 47 | 40 |
| | Julius | 23 | 50 | 27 |
| | Harbor | 87 | 10 | 3 |
| Chl-*a* | Histadrut | 5 | 61 | 34 |
| | Julius | 16 | 55 | 29 |
| | Harbor | 24 | 71 | 5 |

### 3.4. Relationships between HABs and Nutrient Dynamics

Potentially toxic micro-algal species, which can develop into harmful algal blooms (HABs), are occasionally found in the seaward harbor station in the Kishon Estuary, with total cells abundance ranging from ~60 cells $L^{-1}$ to as high as $1.2 \times 10^7$ cells $L^{-1}$ (Figure 6A). The most dominant potentially toxic algal species were the dinoflagellates: *Akashiwo sanguinea*, *Alexandrium* spp., *Dinophysis acuminata*, *Dinophysis caudata*, *Gymnodinium cf catenatum*, *Prorocentrum minimum*, *Heterosigma akashiwo*, and *Karenia brevis*, and the diatoms *Pseudonitzschia* spp. [19,20]. Their abundance shows large temporal variability, with peaks higher than $10^5$ cells $L^{-1}$, which may consider HABs. Nonetheless, a general decreasing trend is observed between 2002 till 2018, and a variable increase since 2019, in line with the time series of $NO_2^- + NO_3^-$ (Figure 4C,D). Similarly, the chl-*a* time series also shows a general decreasing trend between 2002 till 2018 (superimposed by 2 peaks), and a variable increase since 2019 (Figure 6B).

While the relationships between the nutrient concentrations, algal biomass (as chl-*a*), and potentially toxic algal abundance do not follow a simple linkage and may be affected by hydrographic aspects like stratification intensity and water residence time at the estuary, a general decreasing trend for these components is observed during 2002–2018, and an increase since 2019. It is known that increasing nutrient loads, especially of nitrogen compounds (inorganic and organic), may promote the development and persistence of many HABs species, including those routinely found in the Kishon Harbor [35,36]. Our results show that regardless of the reported reduction of anthropogenic nutrient loads via point sources, an increase in nutrient concentrations is observed since 2019 (Figure 4). The latter increase may be attributed to both, unknown diffuse sources and/or hydrographic changes in the water flow and residence time due to the construction of a new Bayport Terminal in Haifa (constructions ended in 2019), limiting water exchange and retaining particulate matter for a longer time at the outlet of the Kishon River. This suggests that the

management of nutrient inputs to such small inland estuaries should consider changes in hydrographic conditions.

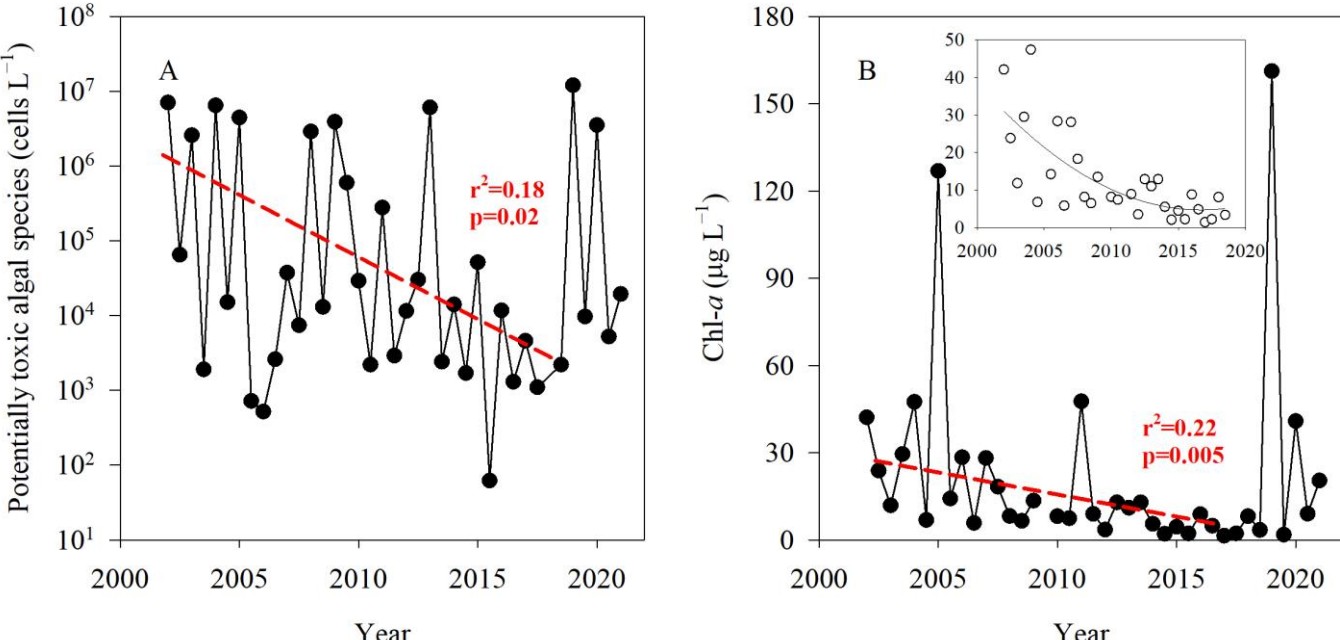

**Figure 6.** Temporal variability of potentially toxic alga abundance (**A**) and chl-*a* (**B**) in the Kishon's Harbor station between 2002–2021. The red lines indicate the decreasing trend in 2002–2018 for both potentially toxic alga (**A**) and chl-*a* (**B**). Insert: chl-*a* levels between 2002–2018 excluding 'abnormal' peaks in 2005 and 2011.

## 4. Summary

Regardless of a certain improvement in nutrient levels recorded over the last 2 decades, the nutrient and chl-*a* levels at the Kishon estuary surface water represent mostly a bad ecological state considering the thresholds adopted in this study. At present, the large variability of nutrient thresholds is used across European countries considering single or multiple elements or nutrient species and applying different methodological/metrics approaches [21]. In addition, the connection of the nutrient thresholds to biological responses or BQE as chl-*a* and HABs is not well established and may be differently linked in each specific water body, especially for small inland stratified estuaries along the Mediterranean coast of Israel. For the Kishon and other small coastal river estuaries, it is suggested to develop a combined suit of variables based on causal relationships between the nutrient and biological variables for assessing GES, mainly chl-*a* and HABs, and considering the relatively high residence time of the estuarine water.

**Author Contributions:** Conceptualization: B.H. and E.R.; Methodology: B.H., Y.G., Y.S., G.S.-V., N.G., N.B. and E.R.; Writing—original draft: B.H. and E.R.; Project administration: B.H. and E.R.; Funding acquisition: B.H. and E.R. All authors have read and agreed to the published version of the manuscript.

**Funding:** This research was funded by the Kishon Water Authority, Haifa, Israel, and the Ministries of Environmental Protection and Energy via the National Monitoring Program of Israel's Mediterranean waters.

**Data Availability Statement:** Data are accessible through the Kishon Water Authority website: www.kishon.org.il/kishon-river-authority.

**Acknowledgments:** The authors are grateful to Danny Lev, Keren Yanuka-Golub, and Racheli Gal from IOLR's marine chemical department for their help over the years in the nutrient measurements.

**Conflicts of Interest:** The authors declare no conflict of interest.

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
