# Peer review of "Long-Term (2002–2021) Trend in Nutrient-Related Pollution at Small Stratified Inland Estuaries, the Kishon SE Mediterranean Case"

_water, doi:10.3390/w15030484_

Round 1

Author Response

Responses to Reviewer 1

We would like to sincerely thank the reviewer for his thoughtful suggestions. We hereafter provide our point-by-point response (blue) to each comment (black) by the reviewer. See attached file.

Line 21: If there Is a residence time calculation and comparison it's not in this manuscript, I would think the residence time is quite long (high) compared to many EU rivers.

Response: Corrected. The text in the discussion refer to relatively lower exchange fluxes and thus higher residence time, and mistakenly in the abstract is written as low.

Line 54: references 10 and 11 are not exactly a source for the fact that our region is governed by small estuaries compared with other regions. If you have a better reference for that please use it.

Response: Corrected. We added a reference for comparison to other areas: Beck et al. 2013 “Global patterns in base flow index and recession based on streamflow observations from 3394 catchments”.

Line 95: Figure 1A – the map is blurred (possibly copy paste quality) and it seems like there is enough space to make it larger.

Response: We tried to improve resolution and added stations and distances in the caption.

Line 102 – please specify the station locations (at least the distance upstream)

Response: Corrected - station names and distances upstream where added to figure 1 caption.

Line 156: you mention the deep water DO but you did not describe sampling there I think the deep water is very interesting but needs a more thorough treatment than anecdotal mention.

Response: We removed the values and refer to references 18 and 20. We focus here on the status and temporal trends of the upper water mass, considering changes in stated nutrient loads (governmental publications) and considering different approaches of water quality criteria. Indeed, the deep water is interesting but evolve additional benthic process which are beyond the scope here.

Figure 3 and the related text (line 179) “we can roughly estimate”: This idea needs some more clarification, maybe paint the two extreme slopes and add their regression line or discuss the range of R2 you are getting for these linear trends per sampling campaign.

Response: The regression lines were added in the text, which is clarified.

Equation 3 (line 243) nut – fw should be nut-fw and same with nut – riv and all other content names. It seems like different phrases than those in previous equations.

Response: corrected

Line 270: consider changing the word variability to time series (in this legend and others). When reading the word "variability" one would expect to see a measure how varible the signal is and not changes of concentration in time

Response: corrected

Figure 4: I looked at the link to the loads, there is no description for the methos of calculation, frequency of measurements and so on. I think the resulting load is highly dependent on the method of calculation, discharge data etc. please describe the method in this manuscript, it will not change the fact that there was a drastic decrease of discharge but will change the details. Also, convert the link to regular citation.

Response: The loads are reported in the web by the Israeli Ministry of Environmental protection, 2021 Report on pollutants budget (in Hebrew). The calculation method is presented in the report text, mainly using the annual cumulative effluent discharge per point source multiplied by the annual average pollutant concentration. While no option for regular citation, we added their calculation method to the figure caption text.

Line 287: Isnt it the other way around? prior to 200 the low ratio indicates N limitation?

Response: mistakenly exchanged, corrected

Figure 5: please indicate the slope of the lines in the figure (or at least in the legend).

Response: added to the figure caption.

Line 304: tranistional should be transitional

Response: Corrected

Line 352: "Which may be considered as HAB" is abundance really the criteria?

Response:  HAB has a complex definition, nonetheless, in some studies, the scale of cells concentration (~above 104) may be used to define a bloom. We therefore use the term “may”.

Line 362: A reduction of seawater exchange rate as suggested in your text will also cause reduced salinity (unless accompanied by reduced freshwater influx) please check it. This comment also relate to the first comment from the abstract (Line 21)

Response: We suggested a potential combined effect of hydrographic conditions (both flows and residence time) and loads. Indeed, when reducing the exchange rate of seawater we may expect lowering the salinity, but we also retain longer and decompose particulate organic matter and enhance nutrient concentrations. To better clarify we added the word ‘flow’ and also added that such reduction of exchange retains particulate matter for longer time at the outlet of the Kishon River.

Line 391: I think the idea of developing a local criteria for the Kishon should be expanded. Why include HAB? They are not a major problem in the Kishon. How is the Kishon different than other estuaries that requires a different criteria?

Response: We suggest using the Kishon as proxy for other small (and similar) coastal river estuaries, not suggesting a specific/local Kishon type criteria. We suggest adding toxic algae measure because such estuaries may serve as a hub for potential toxic algae, as occurred in the Kishon (impacting the estuary and nearby coastal water), see figure 6 and references 20, 24.

Summery: My bottom line: there is one significant thing here, fluxes of fresh water from upstream Kishon should be calculated/presented in this manuscript both to show if there is an increase in residence time since 2019 and to show the changes in nutrient influx since 2000, the source you use is not good enough.

Response: We focus here on the water quality status and temporal trends of the upper water mass, considering changes in nutrient loads (governmental publications) and considering different approaches of water quality criteria.

Reviewer 2 Report

In general, authors have done a decent job preparing this manuscript, which signifies an interesting longer-term study that monitored the nutrient and microalgal concentrations of Kishon River estuary. While there is definitive merit in publishing this manuscript, I have some moderate concerns about the manuscript (see below). I would like the authors to consider the comments below in further improving the manuscript prior to eventual publication. Red-highlighted ones require special attention.

Abstract

L11: use ‘two-decade’ instead of ‘two decades’ & remove the comma

Introduction

L27: ‘enhanced water demands’: try using ‘higher’ (or something likewise) instead of ‘enhanced’. This is because ‘enhanced’ somewhat carries a quality improvement vibe.

L55-56: ‘bottom bathymetry’: isn’t ‘bottom topography’ a better term?

L57: ‘ecological unique’: please use ‘ecologically unique’

L61: consider using ‘two-decade (2002-2021) dataset’

L68: ‘relatively firm nutrient thresholds’ – what does the ‘firm’ mean? – rigorous?

Methods

Line 75: ‘1100 km2’: use ‘approximately 1100 km2’ or ‘ca. 1100 km2’ (ca. = standard abbreviation for Latin term circa)

Figure 1: Unfortunately, this is sub-optimal for publication because the panel A lacks spatial reference. Please add a coordinate grid around the border of the two maps, so users have some spatial reference to the study area. Also, make sure that you make a distinction between water and land (use two color shades or something). This map is otherwise confusing for a reader outside your geographic range.

L81-82: ‘bottom bathymetry’: well, bathymetry is study of beds or floors of water bodies. Therefore, there is no need to say, ‘bottom bathymetry’. Just referring to this as ‘bathymetry’ is sufficient. Make this change throughout the manuscript wherever apparent.

L83: ‘3.8 and 100x106’ this can be easily misleading. Therefore, if you mean to say, ‘3.8 x 106 and 1x108 – mention it likewise (100 x 106 = 1 x 108). Adjust the y-axis scale of Figure 1B also according to this notation (e.g., tickmark label 2 = 2 x 107 and label 7 = 1.2 x 108 and likewise…).

L86: adjust cross reference to ‘Figure 1A’

L89: use ‘primary-producer biomass’

L103: adjust cross reference to ‘Figure 1A’

L104: You sampled in May and October-November. Here you say that this sampling pattern captured the ‘winter/spring’ and ‘summer/autumn’ variability. But how? At ca. 32° N, month of May should be representative of late spring/summer and October-November should be representative of autumn/winter transit, right? There is no way that at such a low latitude, month of May representing a ‘winter/spring’ period. I live at 71° N, and we never call month of May ‘winter or winter/spring’ – it is ‘spring’. Consider this and adjust the seasonal framing accordingly. If not, argue for your choice. If you decide to change the seasonal framing, make those changes throughout the manuscript.

L107: ’10 cm below surface’: I am just curious why only near-surface water samples were collected in this study? Any specific reason behind that choice? In other words, would the results have been different if the samples were taken from mid-water column or near-bottom? Discuss by relating to dissolution and vertical distribution properties of analyzed environmental variables!

L108: for in-situ measurements, from which depth were the measurements taken?

Results & Discussion

L187-188: Table 1 caption: ‘characteristic’: change to ‘characteristics’

Table 1: Please keep spaces between numerals and operators (e.g., 8.3 ± 2.4, NOT 8.3±2.4)

L155: Also comment on how the estimated dispersion (SD in Table 1and range in Figure 2) of many of these variables decrease downstream. It is a very interesting observation, which may indicate a seasonal or interannual effect. Discuss this.

L156: “bottom water low (hypoxic-anoxic) DO concentrations”: Did you measure bottom water for DO? That’s why I asked the question above, ‘which depths were the in-situ measurements of DO were taken?’

L158: ‘DO events’: please replace this term! (e.g., DO measurements)

L161-166: Can you add some of these contrasting mixing curves to an Appendix or Supplement. If you do, add a cross-ref. to that supplement here.

L219-247: Totally out of place. These are methods – which must be placed in the methods section! Please make a subchapter in methods section and move these equations and their descriptions there!

L310: Mention the name or names (et.al suffix) of the authors and then add the indexed citation [32]. It reads well that way.

L325-336: If I am not mistaken, this characterization (nutrient load classification) is based on the mean nutrient estimates of the 20-year period. Since your data show a decreasing trend of nutrient loads in the most upstream location (which should transcend to the rest of the stations in general), wouldn’t it be nice to show how the above characterization (classification) has changed along the timeseries? You may be able to plot this using the same color bands as in Figure 2 on a timeseries plot. See if this can be done. Such an illustration would clearly summarize your main take-home-message. However, a challenge to adopt this type of an illustrative approach is that you use multiple classification schema. So, its up to you to decide whether to adopt this suggestion or not.

General comment: The upward ‘bad’ trend since 2019 is alarming to me. Are any of the authors in discussions with management authorities? Making a scientific publication out of this data is nice but it would be excellent if you can reach out to the Environmental Authority (or equivalent) and communicate these findings on an institutional level.

Author Response

We would like to sincerely thank the reviewer for his thoughtful suggestions. We hereafter provide our point-by-point response (blue) to each comment (black) by the reviewer. Also attached as pdf file.

Abstract

L11: use ‘two-decade’ instead of ‘two decades’ & remove the comma

Response: corrected

Introduction

L27: ‘enhanced water demands’: try using ‘higher’ (or something likewise) instead of ‘enhanced’. This is because ‘enhanced’ somewhat carries a quality improvement vibe.

Response: corrected

L55-56: ‘bottom bathymetry’: isn’t ‘bottom topography’ a better term?

Response: not sure, prefer “bathymetry” for water bodies. This term was used also in Herut et al., 1997 (Marine Pollution Bulletin) and in several underwater depth studies. Generally, Bathymetry is the study of the "beds" or "floors" of water bodies, including the ocean, rivers, streams, and lakes.

L57: ‘ecological unique’: please use ‘ecologically unique’

Response: corrected

L61: consider using ‘two-decade (2002-2021) dataset’

Response: corrected

L68: ‘relatively firm nutrient thresholds’ – what does the ‘firm’ mean? – rigorous?

Response: changed to ‘robust’ as in line 69.

Methods

Line 75: ‘1100 km2’: use ‘approximately 1100 km2’ or ‘ca. 1100 km2’ (ca. = standard abbreviation for Latin term circa)

Response: corrected

Figure 1: Unfortunately, this is sub-optimal for publication because the panel A lacks spatial reference. Please add a coordinate grid around the border of the two maps, so users have some spatial reference to the study area. Also, make sure that you make a distinction between water and land (use two color shades or something). This map is otherwise confusing for a reader outside your geographic range.

Response: Figure revised and grid/coordinates included

L81-82: ‘bottom bathymetry’: well, bathymetry is study of beds or floors of water bodies. Therefore, there is no need to say, ‘bottom bathymetry’. Just referring to this as ‘bathymetry’ is sufficient. Make this change throughout the manuscript wherever apparent.

Response: corrected

L83: ‘3.8 and 100x106’ this can be easily misleading. Therefore, if you mean to say, ‘3.8 x 106 and 1x108 – mention it likewise (100 x 106 = 1 x 108). Adjust the y-axis scale of Figure 1B also according to this notation (e.g., tickmark label 2 = 2 x 107 and label 7 = 1.2 x 108 and likewise…).

Response: corrected (to scientific format)

L86: adjust cross reference to ‘Figure 1A’

Response: corrected

L89: use ‘primary-producer biomass’

Response: corrected

L103: adjust cross reference to ‘Figure 1A’

Response: corrected

L104: You sampled in May and October-November. Here you say that this sampling pattern captured the ‘winter/spring’ and ‘summer/autumn’ variability. But how? At ca. 32° N, month of May should be representative of late spring/summer and October-November should be representative of autumn/winter transit, right? There is no way that at such a low latitude, month of May representing a ‘winter/spring’ period. I live at 71° N, and we never call month of May ‘winter or winter/spring’ – it is ‘spring’. Consider this and adjust the seasonal framing accordingly. If not, argue for your choice. If you decide to change the seasonal framing, make those changes throughout the manuscript.

Response: We referred to the seasonal interfaces (‘late winter-spring’ and ‘late summer-autumn’), which are right but may confuse.

Therefore, we changed as suggested to ‘spring’ and ‘autumn’ (though in certain years it may cover the interface of late/start of seasons).

L107: ’10 cm below surface’: I am just curious why only near-surface water samples were collected in this study? Any specific reason behind that choice? In other words, would the results have been different if the samples were taken from mid-water column or near-bottom? Discuss by relating to dissolution and vertical distribution properties of analyzed environmental variables!

Response: The surface water retains most of the anthropogenic freshwater effluents in this stratified estuary, and we focus here on the status (different approaches of water quality criteria) and temporal trends of the upper water mass, and their responses to changes in stated nutrient loads (governmental publications). The deep water is interesting but evolve much seawater with additional benthic process, which are beyond the scope here.

L108: for in-situ measurements, from which depth were the measurements taken?

Response: added: ~10 cm depth

Results & Discussion

L187-188: Table 1 caption: ‘characteristic’: change to ‘characteristics’

Response: corrected

Table 1: Please keep spaces between numerals and operators (e.g., 8.3 ± 2.4, NOT 8.3±2.4)

Response: corrected

L155: Also comment on how the estimated dispersion (SD in Table 1and range in Figure 2) of many of these variables decrease downstream. It is a very interesting observation, which may indicate a seasonal or interannual effect. Discuss this.

Response: we added a short text on the absolute dispersion while the relative deviations (CV) do not show a distinct general trend.

“The observed scattering (standard deviations; lowest and highest quartiles and data points; Figure 2 and Table 1) of the nutrient and chl-a concentrations decrease downstream, displaying highest variability upstream at the Histadrut station. The latter may reflect fluctuations in the anthropogenic discharges, while the overall variability is also attributed to seasonal imprints.”

L156: “bottom water low (hypoxic-anoxic) DO concentrations”: Did you measure bottom water for DO? That’s why I asked the question above, ‘which depths were the in-situ measurements of DO were taken?’

Response: As mentioned above, we focus here on the status and temporal trends of the upper water mass, considering changes in stated nutrient loads (governmental publications) and their freshwater character and considering different approaches of water quality criteria. The deep water is interesting but evolve much seawater with additional benthic process, which are beyond the scope here.

L158: ‘DO events’: please replace this term! (e.g., DO measurements)

Response: changed to ‘concentrations’

L161-166: Can you add some of these contrasting mixing curves to an Appendix or Supplement. If you do, add a cross-ref. to that supplement here.

L219-247: Totally out of place. These are methods – which must be placed in the methods section! Please make a subchapter in methods section and move these equations and their descriptions there!

Response: The use of the mixing equations to retrieve/assess the anthropogenic end-member is based on the observations presented in the first part of the results, therefore to our opinion properly/rationally placed. Nevertheless, we shifted this part to a new section under the methods upon your request.

L310: Mention the name or names (et.al suffix) of the authors and then add the indexed citation [32]. It reads well that way.

Response: corrected

L325-336: If I am not mistaken, this characterization (nutrient load classification) is based on the mean nutrient estimates of the 20-year period. Since your data show a decreasing trend of nutrient loads in the most upstream location (which should transcend to the rest of the stations in general), wouldn’t it be nice to show how the above characterization (classification) has changed along the timeseries? You may be able to plot this using the same color bands as in Figure 2 on a timeseries plot. See if this can be done. Such an illustration would clearly summarize your main take-home-message. However, a challenge to adopt this type of an illustrative approach is that you use multiple classification schema. So, its up to you to decide whether to adopt this suggestion or not.

Response: Thank you much for suggesting. As noted it is a challenging illustration and we prefer showing the observed concentrations trends and presenting the table with percent of measurements that fall within each category of ecological state.

General comment: The upward ‘bad’ trend since 2019 is alarming to me. Are any of the authors in discussions with management authorities? Making a scientific publication out of this data is nice but it would be excellent if you can reach out to the Environmental Authority (or equivalent) and communicate these findings on an institutional level.

Response: Certainly, we will approach the relevant environmental authorities.

Reviewer 3 Report

Comments in the attachment.

Author Response

Responses to Reviewer 2

We would like to sincerely thank the reviewer for his thoughtful suggestions. We hereafter provide our point-by-point response (blue) to each comment (black) by the reviewer.

The issues contained in the work entitled: "Long-term (2002-2021) trend in nutrient-related pollution at small stratified inland estuaries, the Kishon SE Mediterranean case" are important in the context of the global threat of progressive trophic degradation of inland waters used as drinking water resources as well as the eutrophic threat to coastal waters. Population growth in estuaries and the coastal zone of the Mediterranean Sea entails increased anthropopressure, which, combined with intensifying climate change, may lead to a dangerous water deficit.

In my opinion, the manuscript was drafted relatively correctly. The research problem is well outlined. However, the purpose of the work needs to be clarified. The article is interesting and worth publishing, but it needs some corrections (comments below):

Introduction

  • Lines 61-71: The aim of the work requires clarification.

Response: Clarified, text revised.

Materials and Methods

  • Lines 108-109, 115, 130, 140: Please enter the name of the device manufacturer and country of origin.

Response: Added in the text

  • Line 118 and the rest of the text of the work: Correct the markings: PO4 to PO43-, NO2 to NO2-, NO3 to NO3-, NH4 to NH4+. The use of mineral forms of nitrogen and phosphorus compounds without specifying the ion charge is incorrect and was acceptable at a time when there were no improvements for easy insertion of superscripts.

Response: Corrected, text revised.

  • Line 121: control samples analysis results ?

Response: controls were performed by the CRM as presented in the text.

  • Lines 122-123: „Note that all samples were well above their LOD values.” - In my opinion, this comment is redundant.

Response: Removed.

  • Lines 144-145: "The temporal changes in nutrient and their endmembers, potentially toxic algae, and chl-a were evaluated using a Pearson correlation test ..." - temporal changes or relationships between these parameters?

Response: Text revised to temporal trends (“The temporal trends of nutrient concentrations and their endmembers, potentially toxic algae cell abundance, and chl-a concentrations were evaluated using a Pearson correlation test with a confidence level of 95% (α=0.05)….”)

Results and Discussion

  • Lines 156-157: Where are these results coming from, please provide the source.

Response: We removed the values and refer to references 18 and 20.

  • Lines 162, 165, 170 and more: conservative ? conventional ?

Response: conservative (as in the text)

  • Line 176: nutrient fertilizers ?

Response: revised to “impacted by fertilizers”

  • Tables, Figures and text: why are concentrations of mineral forms of nitrogen and phosphorus given in μmol·L-1, and not mg·L-1, as is commonly used? This makes it difficult to compare values without converting them. DO and chl-a are given in mg·L-1 and μg·L-1 (respectively).

Response: Several marine and estuarine studies presents nutrient concentrations in molar units. It is referred to bio-availability and nutrient limitation assessments (usually in reference to Redfield ratio) and avoiding the potential mishmash between the weight unit of the whole molecule (e.g. NO3-) or the nutrient atom (N). We thus prefer presenting the nutrient concentrations in molar units.

  • Table 1: In the case of pH, an average value is not calculated, but a range is given.

Response: The table contain an average and stdev, which may characterize the water acidity.

  • Line 195: only NO3? the axis in Figure 2C is labeled NO2+NO3.

Response: Corrected

  • Line 202: „Note the different Y axis” - In my opinion, this comment is redundant.

Response: Corrected (removed)

  • Lines 224, 230, 243: The equation number is placed on the right side near the margin in the form of an Arabic numeral in round bracket, e.g. (1), (2), (3)...

Response: Corrected for numbering (next to equation)

  • Equations, especially equation 3, need to be refined in the equation editor (multiplication signs and spaces between pauses).

Response: Corrected

  • Figure 4 (especially A and C) are hard to read. It is impossible to analyze the nitrogen and phosphorus loads presented there.

Response: The loads in tabular form can be retreived from the web link given in the text.

  • Figure 5 (A and B) clearly presents the literature data, while the research data are concentrated in one place and hardly visible.

Response: The main purpose of the figure is to show the significant change between this study and past nutrient ratios, at the same scale space.

  • Line 310: presented by [32].

Response: Corrected

  • Lines 310-312: Limit values for nitrogen and phosphorus are presented in mg·L-1 and μg·L-1 (respectively), while in this paper the μmol·L-1 units were used.

Response: Corrected

  • Table 3. Please check the data in table 3 very carefully, especially for both states (values for good and moderate are the same) according to Poikane et al.,[20] and according to Nikolaidis et al.,[31] (for nitrogen and phosphorus, these values are same).

According to the classification created for these study, the same value also classifies into two different states (moderate and good, or bad in the case of DO).

Response: The table was carefully checked and revised. In reference [20] (and others) a range (and median) of potential threshold values are given for different types of rivers, which cause overlaps, and we tryed to show it. It indeed complicate, therefore, we revised the table to be more simple by presenting only median values. For this study we corrected the sign “≤” for the moderate state.

  • Lines 358, 374: Concentration is not an appropriate term for the amount of algae cells. The correct term is abundance or biomass

Response: we changed the term “cell concentrations” to “abundance” (nonetheless, to the best of our knowledge both are correct).

Reviewer 4 Report

The authors report on a long-term study of nutrient-related pollution in the SE Mediterranean. The manuscript will be of use to researchers and practitioners. The article should be acceptable for publication in water after attention to the following comments.

Line17: In spite “of.”

Line 20: Change “suit” for “suite.”

Keywords: It seems that the keyword “water” is useless because that is the journal title. Exchange for “quality” or “water quality.”

Line 65: What exactly is meant by “non-good” in the context of this sentence?

Line 87: “influences”

Line 113: Volume?

Line 149: “dissolved”

Line 202: “axes”

Lines 260–264: Could it possibly be connected to COVID restrictions/quarantines in which populations shifted and remained home or worked from home? See papers such as Fritsche et al., AWWA Water Science 2022; 4(3) e1286

Line 306: “enabling more large-scale assessments”

Line 313: “presents”

Line 319: choose a word other than “firm,” perhaps you mean “robust?”

Author Response

Responses to Reviewer 3

We would like to sincerely thank the reviewer for his suggestions. We hereafter provide our point-by-point response (blue) to each comment (black) by the reviewer. See attached file.

Line17: In spite “of.”

Response: corrected

Line 20: Change “suit” for “suite.”

Response: corrected

Keywords: It seems that the keyword “water” is useless because that is the journal title. Exchange for “quality” or “water quality.”

Response: corrected

Line 65: What exactly is meant by “non-good” in the context of this sentence?

Response: sentence removed in the revised version

Line 87: “influences”

Response: corrected

Line 113: Volume?

Response: added in the text, triplicates of 20 ml.

Line 149: “dissolved”

Response: added in the text

Line 202: “axes”

Response: sentence/wording removed.

Lines 260–264: Could it possibly be connected to COVID restrictions/quarantines in which populations shifted and remained home or worked from home? See papers such as Fritsche et al., AWWA Water Science 2022; 4(3) e1286

Response: A significant source of the nutrient discharge is via industrial plant effluents, not reporting any change. Thus, it seems too speculative and difficult to support in evidences. We do see impacts on other aspects as coastal litter amounts.

Line 306: “enabling more large-scale assessments”

Response: corrected

Line 313: “presents”

Response: corrected

Line 319: choose a word other than “firm,” perhaps you mean “robust?”

Response: corrected

Round 2

Reviewer 2 Report

I thank the authors for their responses to my comments and changes adopted to the text in turn. I have no further comments at this point, and thus will give a green light to acceptance. 

I wish authors all the best in their future scientific work.